# Furanocoumarins as Enhancers of Antitumor Potential of Sorafenib and LY294002 toward Human Glioma Cells In Vitro

**DOI:** 10.3390/ijms25020759

**Published:** 2024-01-07

**Authors:** Joanna Sumorek-Wiadro, Adrian Zając, Krystyna Skalicka-Woźniak, Wojciech Rzeski, Joanna Jakubowicz-Gil

**Affiliations:** 1Department of Functional Anatomy and Cytobiology, Institute of Biological Sciences, Maria Curie-Skłodowska University, Akademicka 19, 20-033 Lublin, Poland; joanna.sumorek-wiadro@mail.umcs.pl (J.S.-W.); adrian.zajac@umcs.pl (A.Z.); wojciech.rzeski@umcs.pl (W.R.); 2Independent Laboratory of Natural Products Chemistry, Medical University of Lublin, Chodźki 1, 20-093 Lublin, Poland; kskalicka@pharmacognosy.org; 3Department of Medical Biology, Institute of Rural Health, Jaczewskiego 2, 20-950 Lublin, Poland

**Keywords:** furanocoumarins, imperatorin, isoimperatorin, bergapten, xanthotoxin, gliomas, LY294002, sorafenib, programmed cell death, apoptosis

## Abstract

Furanocoumarins are naturally occurring compounds in the plant world, characterized by low molecular weight, simple chemical structure, and high solubility in most organic solvents. Additionally, they have a broad spectrum of activity, and their properties depend on the location and type of attached substituents. Therefore, the aim of our study was to investigate the anticancer activity of furanocoumarins (imperatorin, isoimperatorin, bergapten, and xanthotoxin) in relation to human glioblastoma multiforme (T98G) and anaplastic astrocytoma (MOGGCCM) cell lines. The tested compounds were used for the first time in combination with LY294002 (PI3K inhibitor) and sorafenib (Raf inhibitor). Apoptosis, autophagy, and necrosis were identified microscopically after straining with Hoechst 33342, acridine orange, and propidium iodide, respectively. The levels of caspase 3 and Beclin 1 were estimated by immunoblotting and for the blocking of Raf and PI3K kinases, the transfection with specific siRNA was used. The scratch test was used to assess the migration potential of glioma cells. Our studies showed that the anticancer activity of furanocoumarins strictly depended on the presence, type, and location of substituents. The obtained results suggest that achieving higher pro-apoptotic activity is determined by the presence of an isoprenyl moiety at the C8 position of the coumarin skeleton. In both anaplastic astrocytoma and glioblastoma, imperatorin was the most effective in induction apoptosis. Furthermore, the usage of imperatorin, alone and in combination with sorafenib or LY294002, decreased the migratory potential of MOGGCCM and T98G cells.

## 1. Introduction

Gliomas are the most common tumors of the central nervous system and constitute approx. 30–40% of intracranial tumors. The World Health Organization, based on the histological assessment of malignancy, has divided gliomas using a four-level scale (I–IV). Grade I and II tumors are characterized by a low degree of malignancy, and surgical intervention in most cases allows for complete recovery. In turn, grades III and IV are infiltrating and vascularized gliomas with a high degree of invasiveness, and the average survival time of patients does not exceed five and two years, respectively, from the moment of diagnosis [1,2]. Among them, the most aggressive are anaplastic astrocytoma (AA, III^O^ WHO) and glioblastoma multiforme (IV^O^ WHO). Their cancerous transformation is associated with the occurrence of mutations in genes whose products are engaged in signal transmission from the cell membrane to the nucleus. In gliomas, the consequences of the incorrect expression of these genes is the activation of key signaling pathways responsible for cell survival: RAS-RAF-MEK-ERK (RAS—Ras protein; RAF—rapidly accelerated fibrosarcoma; MEK—mitogen-activated protein kinase; ERK—extracellular signal-regulated kinase) and PI3K-AKT/PKB-mTOR (PI3K—phosphoinositide 3-kinase; AKT/PKB—protein kinase B; mTOR—mammalian target of rapamycin kinase). The excessive survival signal transduction occurring in gliomas is attributed to the overexpression of genes encoding EGFR (endothelial growth factor receptor) and PDGFR (platelet-derived growth factor receptor) receptors. More than half of GBM patients have abnormalities in the EGFR gene in the form of its amplification or EGFRvIII mutation (EGFR variant III), leading to ligand-independent signaling. The consequence of this is increased AKT/PKB or RAF activity, observed in 80% and 50% of gliomas, respectively [3].

Furanocoumarins are two-ring derivatives of benzo-α-pyrone that have a furan ring substituted at the C6-C7 or C7-C8 position (Figure 1).

These naturally occurring compounds in the plant world, belonging to the class of secondary metabolites, are characterized by low molecular weight, simple chemical structure, and high solubility in most organic solvents. Additionally, they have a broad spectrum of activity, and their properties depend on the location and type of attached substituents. Thanks to these features, they are potential candidates in the construction of new therapeutic strategies [4]. Their cytotoxic effect on transformed cells has been demonstrated. Furanocoumarins inhibit proliferation and limit angiogenesis and metastasis, so they can be used in chemotherapy [4,5,6,7]. It has been proven that the anticancer activity of furanocoumarins is related to the presence of a methoxy or isoprenyl substituent in their structure, which can be attached to the C5 or C8 position of the coumarin ring. In the case of methoxy derivatives, these are bergapten (5-MOP—5-methoxypsoralen) and xanthotoxin (8-MOP—8-methoxypsorelen), while the isoprenyl substituent is attached to isoimperatorin and imperatorin, respectively. Both bergapten and xanthotoxin have pro-apoptotic and anti-proliferative properties and inhibit the cell cycle in the G1 phase in the human leukemia cell line (HL60) [8]. 5-MOP also inhibits the development of liver cancer (HepG2) and stomach cancer (SGC-7901) [9]. It has also been shown that the pro-apoptotic activity of the compound is related to its ability to inhibit PI3K kinase activity in MCF-7 and ZR-75 breast cancer cells [10,11,12]. In the case of the bergapten isomer—xanthotoxin—the induction of apoptosis in HepG2 cells is associated with the inhibition of survival signals through the intracellular RAS-RAF-MEK-ERK pathway [13,14]. Another furanocoumarin with defined biological activity is imperatorin. Its anticancer effects have been observed in many types of cancer, including: leukemias, cervical cancer, gliomas, and liver cancer, and this activity is associated with downregulation of the PI3k-Akt/PKB pathway (SGC-7901 gastric cancer cells) [15,16,17]. In turn, the isomer of imperatorin—isoimperatorin–induces apoptosis in prostate cancer cells (DU145) and gastric cancer cells (SGC-7901) [18].

Therefore, previous studies have suggested that the anticancer activity of the tested furanocoumarins may be related to the inhibition of signal transduction via the RAS-RAF-MEK-ERK and PI3K-AKT/PKB mTOR pathways [7]. It is known from the literature that the use of inhibitors of proteins involved in signal transduction through the mentioned signaling pathways significantly increases the sensitivity of glioma cells to the induction of apoptosis [19,20]. Therefore, this study attempted to determine the role of furanocoumarins as adjuvants in the induction of programmed death in combination with the RAF kinase inhibitor—sorafenib and the PI3K kinase inhibitor—LY294002.

## 2. Results

### 2.1. Induction of Programmed Cell Death by Furanocoumarins: Bergapten, Xanthotoxin, Imperatorin, and Isoimperatorin

MOGGCCM and T98G cells were incubated for 24 h with furanocoumarins: bergapten, xanthotoxin, imperatorin, and isoimperatorin in final concentrations: 25, 50, 100, and 150 µM. The results proved that the effectiveness of the used compounds in induction programmed cell death depended on both their concentration and cell line (Figure 2 and Figure 3).

In the MOGGCCM cell line, the most effective inducer of apoptosis was imperatorin. This compound at a concentration of 50 µM induced apoptosis in approx. 17% of cells, while higher concentrations (≥100 µM) also caused necrosis (Figure 2c). The anticancer activity of other compounds was not significant, and the percentage of apoptotic cells did not exceed 6% (Figure 2a,b,d). As in the case of anaplastic astrocytoma, and also in glioblastoma multiforme, imperatorin proved to be the most effective. Furanocoumarin at a concentration of 50 µM and 100 µM initiated the apoptosis in 17% and 25%, respectively (Figure 3c). However, higher concentrations of the compound (≥100 µM) were also accompanied by necrosis (about 10%). Xanthotoxin at concentrations ≥100 µM showed high effectiveness in inducing apoptosis too, but after application of 150 µM of the compound, additionally necrosis was observed (about 10%) (Figure 3b). Xanthotoxin and imperatorin isomers: bergapten and isoimperatorin, respectively, had no significant effect on the induction of programmed cell death in the T98G line. Studied compounds had no impact on autophagy initiation in both studied cell lines, and at a concentration of 150 µM additionally induced necrosis. For this reason, a final concentration of furanocoumarins of 50 µM was used in further studies.

### 2.2. Effect of Furanocoumarins (Bergapten, Xanthotoxin, Imperatorin and Isoimperatorin) in Combination with Sorafenib and LY294002 on Programmed Cell Death Induction

It is known that the inhibitors of proteins involved in signal transduction by RAS-RAF-MEK-ERK and PI3K-AKT/PKB-mTOR signaling pathways significantly increases the sensitivity of glioma cells to the induction of apoptosis [19,20] Therefore, in addition to a single application of furanocoumarins, they were combined with the RAF kinase inhibitor—sorafenib and the PI3K kinase inhibitor—LY294002 and the microscopic observations of characteristic morphological changes were conducted (Figure 4). The final concentrations of furanocoumarins, sorafenib and LY294002 used in the studies were 50 µM, 10 µM, and 1 µM, respectively.

#### 2.2.1. Furanocoumarins in Combination with Sorafenib

In the MOGGCCM line, sorafenib had no impact on the level of cell death, which was comparable to the control one. Co-incubation of the drug with tested furanocoumarins showed no significant effect on the pro-apoptotic properties of bergapten, xanthotoxin, and isoimperatorin (Figure 5a). However, a significant decrease was observed in the number of apoptotic cells in comparison to imperatorin alone after the simultaneous application of sorafenib and furanocoumarin, and the antagonism of the effect was confirmed using the Chou–Talalay test. Both compounds did not initiate autophagy or necrosis. Different results were obtained in T98G. Sorafenib appeared to be an effective autophagy inducer (12%), having no effect on apoptosis and necrosis (Figure 5b). The simultaneous application of sorafenib and furanocoumarins inhibited the pro-autophagic properties of the RAF kinase inhibitor. Moreover, imperatorin in combination with sorafenib caused a synergistic effect in the induction of apoptosis, initiating this process in approx. 24% of cases. In contrast to the imperatorin, the pro-apoptotic potential of its isomer—isoimperatorin and bergapten in combination with sorafenib—did not change compared to a single application of furanocoumarins. The antagonistic effect was observed when sorafenib was combined with xanthotoxin, and the apoptotic properties of furanocoumarin decreased by approx. 10%.

#### 2.2.2. Furanocoumarins in Combination with LY294002

Microscopic analysis of anaplastic astrocytoma and glioblastoma cells incubated with LY294002 showed inhibitor-initiated autophagy in approx. 43% of MOGGCCM cells and almost 20% of T98G cells (Figure 6a,b). Additionally, apoptosis was also observed in 9% of the AA and 28% of the GBM. The usage of the tested furanocoumarins in simultaneous application with LY294002 completely reduced the sensitivity of tumor cells to autophagy induced by LY294002. Moreover, in the T98G line, the use of a combination therapy was associated with a decrease in apoptotic cells, compared to incubation with only LY294002. Different effects were obtained in the MOGGCCM line, where simultaneous incubation with LY294002 and imperatorin increased the number of apoptotic cells (16%), compared to the application of the PI3K inhibitor alone. However, the number of dead cells using this method of application was comparable to the level observed after the application of only furanocoumarin (16%). Furanocoumarins, alone and in combination with sorafenib or LY294002, were not cytotoxic to the normal cells—the OLN-93 oligodendrocyte cell line (Figure 5c and Figure 6c).

### 2.3. Estimation of the Level of Beclin 1 and Procaspase 3 and Activity of Caspase 3

At the molecular level, cell death processes are regulated by specific marker proteins. Caspase 3 plays a key role in apoptosis, while autophagy is regulated by Beclin 1. The results obtained after incubation of MOGGCCM and T98G cells with LY294002 showed that the inhibitor significantly increased the level of Beclin 1 in both cell lines. In the case of sorafenib, the compound did not change the amount of Beclin 1 in AA cells, but did increase its level in GBM cells. The application of imperatorin provided the opposite results. Furanocoumarin only increased the level of Beclin 1 in anaplastic astrocytoma. Interestingly, the usage of a combination of imperatorin with both sorafenib and LY294002 significantly reduced the level of the autophagy marker in both cell lines by about 40% (Figure 7c,d). Quantitative and qualitative analyses of the immunoblots revealed that sorafenib and imperatorin, applied alone, decreased the level of procaspase 3 in the MOGGCCM cell line; meanwhile, in T98G, the level of this protein was lower upon simultaneous application of either both compounds or only LY294002 (Figure 7a,b). In the induction of apoptosis, the presence of the active form of the caspase 3 plays a key role. According to our study, imperatorin increased caspase 3 activity in both cell lines. Furthermore, in the MOGGCCM and T98G cell lines, similar effects were obtained after application of furanocoumarin with LY294002 or sorafenib, respectively.

### 2.4. Apoptosis, Autophagy, and Necrosis Induction upon Inhibition of RAF and PI3K Expression

In order to confirm the involvement of the RAS-RAF-MEK-ERK and PI3K-AKT/PKB mTOR pathways in the resistance of gliomas to the induction of programmed cell death, the first elements of these signaling pathways, i.e., RAF and PI3K, were blocked with specific siRNA (Figure 8). The conducted research showed that blocking of the expression of PI3K and RAF kinases was more effective than the use of inhibitors of these proteins, i.e., LY294002 and sorafenib, respectively, which resulted in greater sensitivity to apoptosis induction using imperatorin. Interestingly, autophagy and necrosis did not occur in cells with blocked RAF or PI3K expression. In siRAF-1-transfected cells, furanocoumarin induced apoptosis by almost 100% in both cell lines. Similar effects were obtained after the simultaneous application of imperatorin and LY294002 in the MOGGCCM line; meanwhile, in T98G, the number of apoptotic cells did not exceed 70%. Application of LY294002 alone was less effective and induced apoptosis in approx. 60% of AA cells and 85% of GBM cells. In the case of T98G cells transfected with siPI3K, both imperatorin and sorafenib, in single and simultaneous application, induced apoptosis in over 90% of cells. The AA cells with the blocked expression of PI3K were less sensitive to induction of programmed cell death upon imperatorin (40%), alone and in combination with LY294002 treatment (70%).

### 2.5. Effect of Imperatorin, Alone and in Combination with Sorafenib or LY294002 on the Migration Potential

Gliomas are characterized by a high degree of invasiveness and metastasis. Therefore, the effect of imperatorin, in single and simultaneous application with sorafenib or LY294002, on the migration potential of gliomas was examined by the scratch test (Figure 9 and Figure 10). Sorafenib and LY294002, in single application, showed similar activity and reduced the migration potential of AA cells by approx. 60% (Figure 9).

The best results were obtained following imperatorin treatment alone (reduced 80%), while the combined application with sorafenib or LY294002 reduced the ability of furanocoumarin to inhibit the mobility of anaplastic astrocytoma cells. In the T98G cell line, LY294002 reduced the migration potential to a similar extent to the MOGGCCM, which was 60% (Figure 10). Sorafenib turned out to be less effective and reduced migration by approx. 40%. The most effective was imperatorin, alone and in combination with sorafenib or LY294002, which reduced migration potential by 65%.

## 3. Discussion

Gliomas are tumors of the central nervous system that have an extremely poor prognosis. Currently used treatments only allow an extension of the patient’s life and improvement of its quality. Despite surgical resection and the use of adjuvant radio–chemotherapy, they are characterized by rapid tumor recurrence, which is why new drugs are still being sought. Considering their widespread availability and broad spectrum of action, furanocoumarins are of immense importance. It has been shown that the use of these compounds with chemotherapy increases the anticancer effect of cytostatics. Their chemical structure has a profound influence on the anticancer properties. According to the data obtained from the literature, anticancer activity is possessed by methoxy (bergapten and xanthotoxin) and isopentenyloxy derivatives (isoimperatorin and imperatorin), substituted at the C5 (bergapten, isoimperatorin) and C8 (xanthotoxin, imperatorin), respectively.

Our studies showed that the anticancer activity of furanocoumarins strictly depended on the presence, type, and location of substituents. In both anaplastic astrocytoma and glioblastoma, imperatorin was the most effective, inducing apoptosis initiation in approx. 17% of cells. At the molecular level, it was accompanied by an increase in caspase 3 activity. Moreover, our research shows that imperatorin also had the ability to reduce the migration potential of gliomas III^O^ and IV^O^, which may have great application significance. It is known that the high mortality and difficulties in treating these tumors are related to their potential for rapid, infiltrative growth and the formation of distant metastases [21,22]. The high mobility group box 2 protein (HMGB2) may play a key role in this process, responsible for the increased expression of cyclin A and genes involved in the epithelial–mesenchymal transition process. As the latest research shows, imperatorin has the ability to reduce the activity of this protein [23].

As already mentioned, the anticancer properties of furanocoumarins can be considered based on their chemical structure. Imperatorin has an isopentenyl substituent at the C8 position attached to the coumarin ring via an oxygen atom. Our previous studies have shown that the presence of this substituent also increases the ability of simple coumarin to induce apoptosis in MOGGCCM and T98G cells [24]. Interestingly, in addition to isoprenyl alkyl substitution, its location plays a significant role in the anticancer activity of the compound. In fact, the incubation of AA and GBM cells with isoimperatorin was less effective in inducing programmed cell death compared to incubation with its isomer—imperatorin. The obtained results suggest that achieving higher pro-apoptotic activity is determined by the presence of an isoprenyl moiety at the C8 position of the coumarin skeleton, which is additionally confirmed by the results obtained in breast cancer cells (MCF-7 line) [25]. In the case of HeLa cells, changing the location of the isoprenyl substituent from position C5 to C8 did not affect the properties of the compound [26]. Therefore, it can be concluded that, apart from the place of attachment of the substituent, the type of cell line used plays a key role.

Moreover, the data obtained from the literature indicate that replacing the isoprenyl moiety of imperatorin or isoimperatorin with the methoxy substituent of bergapten or xanthotoxin significantly reduces the sensitivity of the HeLa cell line to the induction of apoptosis [26]. These reports are consistent with our results obtained after the application of imperatorin and xanthotoxin in AA and GBM. The methoxy derivative (xanthotoxin), substituted at the C8 position, was less effective than the isoprenyl derivative (imperatorin), in which the substituent was located in the same place. Comparable results were obtained by Sigurdsson et al., who observed that the antiproliferative activity of imperatorin against pancreatic cancer cells (Panc-1 line) significantly exceeded that of xanthotoxin [27]. In the case of compounds containing a moiety at the C5 position (bergapten and isoimperatorin), their properties did not depend on the type of attached substituent (methoxy and isoprenyl), and the level of apoptosis induction was similar. Moreover, our studies showed that T98G cells were more sensitive to the effects of xanthotoxin than bergapten, which was also observed by Widelski et al. in leukemia cells (HL-60) [7]. Such a correlation did not occur in the MOGGCCM cell line, where both coumarins acted at a similar level.

Since the RAS-RAF-MEK-ERK and PI3K-AKT/PKB-mTOR pathways exhibit excessive activity (which increases the resistance of gliomas to pharmacotherapy), the molecularly targeted therapy, which targets the transmitter proteins of these signaling pathways, appears to be a promising treatment option [28]. Therefore, in addition to a single application of the furanocoumarins, we used them in combination with inhibitors of these proteins: sorafenib (RAF inhibitor) and LY294002 (PI3K inhibitor).

Sorafenib, a drug currently used to treat renal cell carcinoma and hepatocellular carcinoma, as well as thyroid cancer, is an inhibitor of RAF kinases. Latest reports indicate that the compound inhibits the formation of new blood vessels and the proliferation of glioma cells [29,30]. Our studies showed that the pro-apoptotic effect of sorafenib in relation to the MOGGCCM and T98G lines was insignificant; however, in T98G cells, the compound effectively induced autophagy (12%), which was associated with an increase in the level of Beclin 1. Our previous studies have shown that sorafenib is an effective adjuvant of both simple coumarins and flavonoids; therefore, its combination with furanocoumarins was used in our studies [31,32,33]. It turned out that the combination of compounds significantly reduced the pro-autophagic activity of the RAF inhibitor, which was associated with a decrease in the level of Beclin 1. In the T98G line, we also noted a much higher pro-apoptotic effectiveness of the simultaneous application of sorafenib with imperatorin, which can be explained by a decrease in the level of procaspase 3 and an increase in the activity of caspase 3. Our previous studies also showed that sorafenib significantly reduced RAF kinase levels [34]. It is known that RAF, through its ability to activate proteins from the IAP family (inhibitors of apoptosis), responsible for inactivating caspases, inhibits apoptosis. Moreover, the kinase has the ability to phosphorylate Bcl-2 family proteins, including the pro-apoptotic Bad protein. Then, the protein located in the cytosol does not create megachannels in the mitochondrial membrane through which cytochrome c is released [35]. Interestingly, our studies showed that anaplastic astrocytoma cells were more resistant to induction-programmed cell death by sorafenib in combination with furanocoumarins compared to glioblastoma multiforme. The obtained results indicate that combination therapy using sorafenib may be more effective in gliomas with higher stages of malignancy, which is consistent with previous reports, according to which the use of an inhibitor simultaneously with quercetin, lensoside Aβ, and LY294002 was more effective in GBM compared to AA [31,32,33,34].

LY294002 is a morpholine derivative of quercetin with strong anticancer properties. It has been shown that this compound reduces the invasiveness of U87 glioma cells (GBM) by diminishing the level of PI3K and AKT/PKB activity [36]. In our studies, the compound induced autophagy in approx. 40% of anaplastic astrocytoma cells, which correlated with an increase in the level of Beclin 1. In the case of glioblastoma multiforme, this process occurred in approx. 20% of cells, and the dominant type of death was apoptosis. At the molecular level, there was a renewed decrease in the level of procaspase 3. Our previous studies have shown that this process is accompanied by a decrease in the mitochondrial membrane potential and an increase in the activity of caspases 3 and 9, which may suggest that apoptosis is induced via the mitochondrial pathway [37]. In our studies, we showed that the autophagic properties of the PI3K inhibitor were eliminated by the simultaneous application of furanocoumarins, which was accompanied by a decrease in the level of Beclin 1. At the same time, we noted that such therapy is less effective than a single application of most furanocoumarins, except imperatorin. In the case of the MOGGCCM cell line, the combination of imperatorin and sorafenib induced apoptosis in approx. 17% of cells, which was associated with an increase in caspase 3 activity. In T98G cells, the effectiveness of simultaneous incubation was akin to imperatorin treatment alone.

As we have shown, both sorafenib and LY294002 have the ability to reduce the mobility of cancer cells. It has been described that sorafenib inhibits the expression of extracellular matrix proteins—MMP-1 and MMP-9—and increases the level of the adhesive protein—E-cadherin—in breast cancer cells (MCF-7 and MDA-MB-231) [38]. In turn, the activity of LY294002 is associated with the inhibition of the expression of metalloproteinase 2, as well as an increase in the level of early growth response factor 1 (EGR-1). The protein is responsible for regulating the expression of genes encoding proteins involved in carcinogenesis [39,40]. Interestingly, the conducted research showed that in the T98G line, simultaneous incubation with imperatorin was more effective than the application of inhibitors alone, while in the MOGGCCM line, better effects were achieved using monotherapy.

To obtain direct evidence for the involvement of the RAS-RAF-MEK-ERK and PI3K-AKT/PKB-mTOR pathways in increasing the resistance of gliomas to the induction of programmed cell death by chemotherapeutics, we blocked the expression of RAF and PI3K kinases by specific siRNAs. The conducted research showed that silencing the expression of both proteins significantly increased the pro-apoptotic activity of sorafenib, LY294002 and imperatorin, both in single and combined application. In the case of anaplastic astrocytoma, silencing the expression of RAF kinase was more effective; meanwhile, in glioblastoma cells, the silencing of PI3K kinase was a better option. Then, the application of imperatorin almost eliminated glioma cells via apoptosis (>90%). The obtained results indicate the enormous potential of inhibiting intracellular conductance using the mentioned pathways in increasing the sensitivity of glioma cells to the induction of apoptosis.

## 4. Materials and Methods

### 4.1. Isolation of Furanocoumarins

In parallel to the previously reported [24] isolation of coumarins, furanocoumarins were also isolated from plant materials. Once parsnip (*Pastinaca sativa* L.) and hogweed (*Heracleum leskovii* Grossh) fruits were extracted using dichloromethane, xanthotoxin and bergapten were separated, and imperatorin was obtained from the methanolic extract of angelica (*Angelica officinalis*). The extraction of isoimperatorin was achieved by first using ethanol and then ethyl acetate on the roots of angelica (*Angelica dehurica*), following a period of drying. After that, the extracted materials were separated using a semi-preparative column (capacity 137 mL) and a high-performance Spectrum countercurrent chromatograph (Dynamic Extractions). Using n-heptane, ethyl acetate, methanol, and water in volume proportions of 1:1:1:1 for xanthotoxin and imperatorin, 5:5:4:6 for isoimperatorin, and 6:5:6:5 for bergapten, the two-phase system was established. HPLC-DAD analyses were used to determine the separated compounds, and the purity was greater than 98%.

### 4.2. Cells and Culture Conditions

A 3:1 mixture of Dulbecco’s modified Eagle medium (DMEM) and Ham’s nutrient mixture F-12 (Sigma, St. Louis, MO, USA), supplemented with 10% fetal bovine serum (FBS, Sigma, St. Louis, MO, USA), and penicillin (100 units/mL)—streptomycin (100 µg/mL) solution (Sigma) was used to grow human anaplastic astrocytoma (MOGGCCM; European Collection of Cell Cultures, ECACC, Porton Down, Salisbury, UK ) and human glioblastoma multiforme cells (T98G; European Collection of Cell Cultures, ECACC, Porton Down, Salisbury, UK). The 1:1 mixture of DMEM and F-12 medium with 10% of FBS, penicillin (100 units/mL; Sigma, St. Louis, MO, USA), and streptomycin (100 µg/mL; Sigma, St. Louis, MO, USA) was used to culture a permanent rat oligodendrocyte cell line (OLN-93), which was donated by the Department of Neonatology, Charite, Campus Virchow Klinikum, Humboldt University, Berlin. A humidified environment (95% air and 5% CO_2_) and 37 °C were used to maintain the cultures.

### 4.3. Drug Treatment

Furanocoumarins as well as LY294002 (PI3K inhibitor; Sigma, St. Louis, MO, USA) and sorafenib (Raf inhibitor; Nexavar; BAY 43–9006) were dissolved in DMSO and the final concentration of the solvent did not exceed 0.01% both in control and treated cells. Furanocoumarin concentrations were chosen experimentally and inhibitor doses were based on previous studies [33,34] and applied to the culture medium of MOGGCCM and T98G cells for 24 h.

### 4.4. Apoptosis, Autophagy, and Necrosis Identification and Level Evaluation

The analysis of type and level of cellular death in the control and tested cultures (OLN-93, MOGGCCM, and T98G cells) was evaluated using fluorochrome staining, Hoechst 33342 (Sigma, St. Louis, MO, USA) and propidium iodide (Sigma, St. Louis, MO, USA) mix for apoptosis and necrosis and orange acridine for autophagy were used according to our previous studies [34]. Using a confocal microscope (Axiovert 200 M with scanning head LSM 5 PASCAL; Zeiss, Jena, Germany), dead cells were analyzed morphologically. Cells with pink fluorescent nuclei were considered necrotic, whereas those with blue fluorescent nuclei (fragmented and/or with condensed chromatin) were considered apoptotic. By studying the distinctive acidic vesicular organelles (AVOs) that are generated during the process of the autophagy, its level was assessed. At least 1000 cells were counted in randomly chosen microscopic regions in triplicate.

### 4.5. Immunoblotting

Control and tested cells were lysed to create whole cell extracts as described in our previous research [34]. A total of 80 μg of proteins was electroblotted onto an Immobilon-P PVDF membrane (Thermo Scientific, Rockford, IL, USA), after being separated by 10% SDS-PAGE. After blocking the membranes with 5% low-fat milk for 30 min, they were incubated with the primary-mice monoclonal antibodies against Beclin 1, and procaspase 3 (0.5 μg/mL; Santa Cruz Biotechnology, Dallas, TX, USA) within an entire night at 4 °C. Next day, after washing them 3 x in PBS buffer with 0.05% Triton X-100 (Sigma), membranes were incubated with secondary, alkaline phosphatase (AP)-conjugated, antibodies. The NBT/BCIP Solution (ABCAM, Waltham, MA, USA) was used to detect proteins. The acquired results were subjected to a qualitative analysis based on the color depth, band width, and band thickness. ImageJ Software version 1.8.0 was used to analyze protein bands quantitatively. The information was adjusted in relation to β-actin (0.5 μg/mL; Santa Cruz Biotechnology). There were three separate experiments carried out. As Appendix A, whole blot membranes are attached (Appendix A).

### 4.6. Caspase 3 Activity Assay

In accordance with our earlier research [34], the activity of caspase 3 in treated cells was assessed using a SensoLyte^®^AMC Caspase Substrate Sampler Kit (AnaSpec, Fremont, CA, USA). An 800 TS microplate reader from BioTek, Santa Clara, CA, USA, with 96-well black microplates were used to analyze the fluorescence of 7-aminocoumarin (AMC) at Ex/Em = 354/422 nm.

### 4.7. MOGGCCM and T98G Transfection with siRNA

Following a 24-h incubation of cells at 2 × 10^5^ density, they were exposed to treatment pursuant with our previous studies [34]. After being washed with a 3:1 DMEM/Ham’s F-12 mixture free of serum and antibiotics, the medium was aspirated. Subsequently, the cells were transfected using 2 μL of transfection reagent (Santa Cruz Biotechnology, Dallas, TX, USA) alongside 2 μL of siRNA (siPI3K and siRAF-1). After five hours at 37 °C, cell cultures were supplemented with a medium containing 20% FBS and a double dose of antibiotics and an additional eighteen hours of incubation was performed. Once the media was changed for a fresh one, the transfected cells were utilized for further investigation. Control siRNA (Fluorescein Conjugates-A; Santa Cruz Biotechnology, Dallas, TX, USA) was used for transfection efficiency monitoring during fluorescence analysis (photos attached in the Appendix A).

### 4.8. Statistical Analysis

GraphPad Prism 5 (GraphPad Software Inc., San Diego, CA, USA) was utilized for statistical evaluation. A one-way ANOVA test and Dunnett’s multiple comparison analysis were the next steps in the process. When analyzing data presented as mean ± standard deviation (SD), the significance threshold was set at *p* < 0.05. Each experiment was performed in triplicate. Chou–Talalay’s test was applied, as in previous research, to confirm the best dose selection for the substances under investigation (Appendix A).

## 5. Conclusions

The obtained results suggest that the anticancer activity of furanocoumarins depends on their chemical structure and cell line. The higher activity is due to the presence of the isoprenyl substituent of imperatorin at the C8 position, and changing its location reduces the pro-apoptotic properties of the compound. In addition to the ability to induce programmed cell death, furanocoumarins also inhibit the migration potential of glioma cells. As we have shown, imperatorin can be used as an adjuvant, increasing the effectiveness of sorafenib and LY294002, and its effectiveness depends on the activity of transmission through the RAS-RAF-MEK-ERK and PI3K-AKT/PKB-mTOR pathways. The obtained results may be the basis for the development of new therapeutic strategies that sensitize glioma cells to the induction of apoptosis.

## Figures and Tables

**Figure 1 ijms-25-00759-f001:**
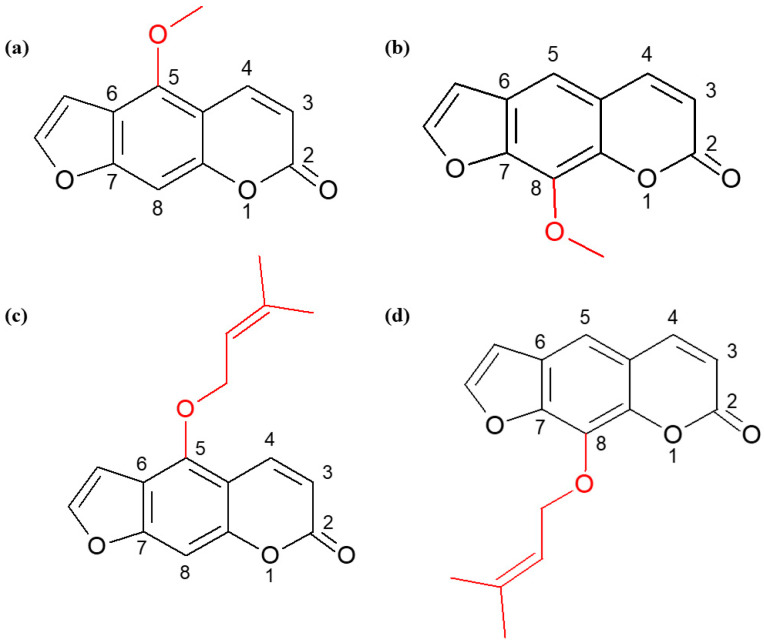
Structural formulas of bergapten (**a**), xanthotoxin (**b**), isoimperatorin (**c**), and imperatorin (**d**).

**Figure 2 ijms-25-00759-f002:**
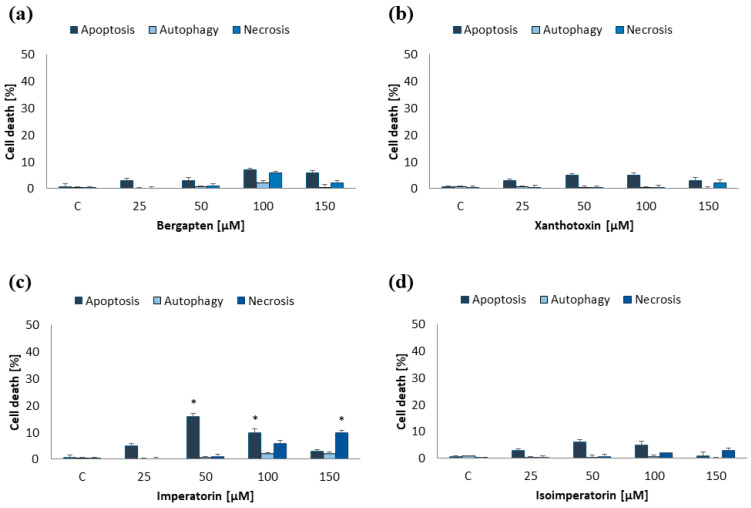
Type and level of cell death observed in MOGGCCM cells after 24 h incubation with different concentrations of bergapten (**a**), xanthotoxin (**b**), imperatorin (**c**), and isoimperatorin (**d**). * *p* < 0.05 compared to control.

**Figure 3 ijms-25-00759-f003:**
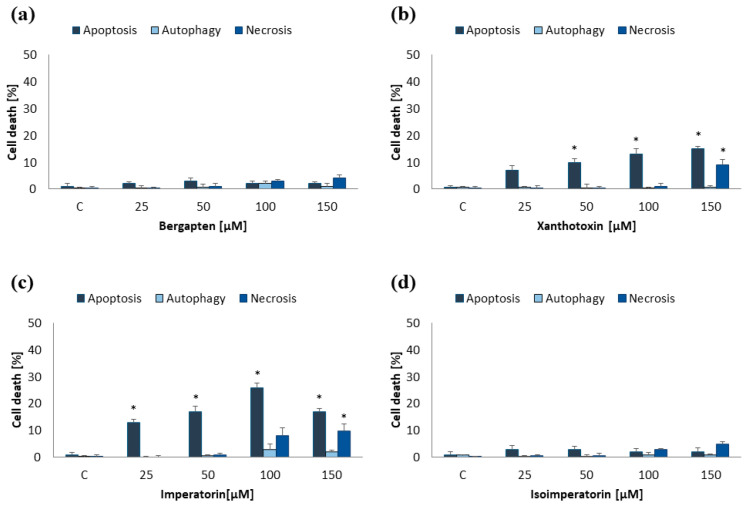
Type and level of cell death observed in T98G cells after 24 h incubation with different concentrations of bergapten (**a**), xanthotoxin (**b**), imperatorin (**c**), and isoimperatorin (**d**). * *p* < 0.05 compared to control.

**Figure 4 ijms-25-00759-f004:**
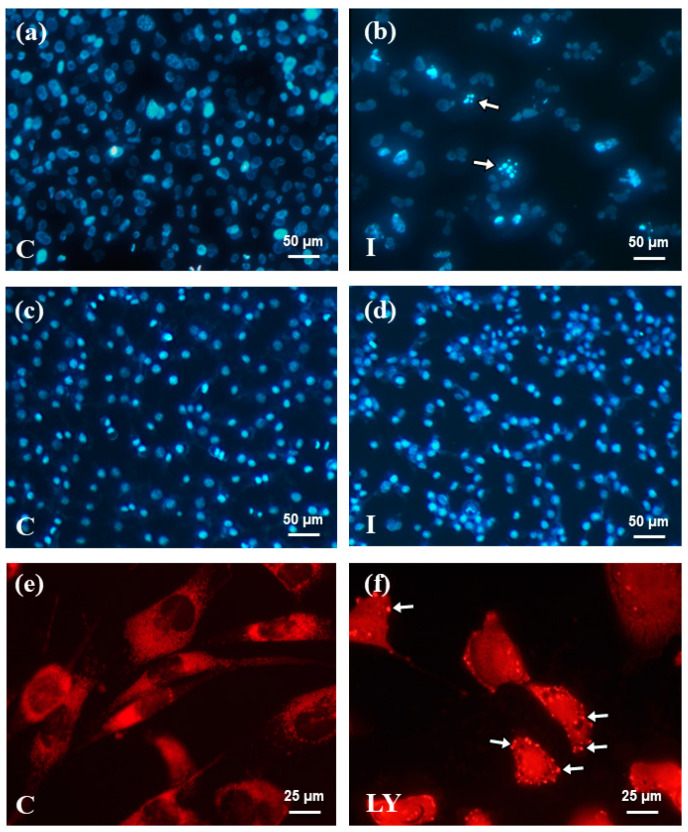
Representative photos of cells stained with T98G (**a**,**b**,**e**,**f**), OLN-93 (**c**,**d**), Hoechst 33342 and propidium iodide (**a**,**d**), and orange acridine (**e**,**f**). Apoptosis (**b**) with characteristic apoptotic bodies (white arrows) and autophagy (**f**) with AVOs (white arrows). C—control; I—imperatorin; LY—LY294002.

**Figure 5 ijms-25-00759-f005:**
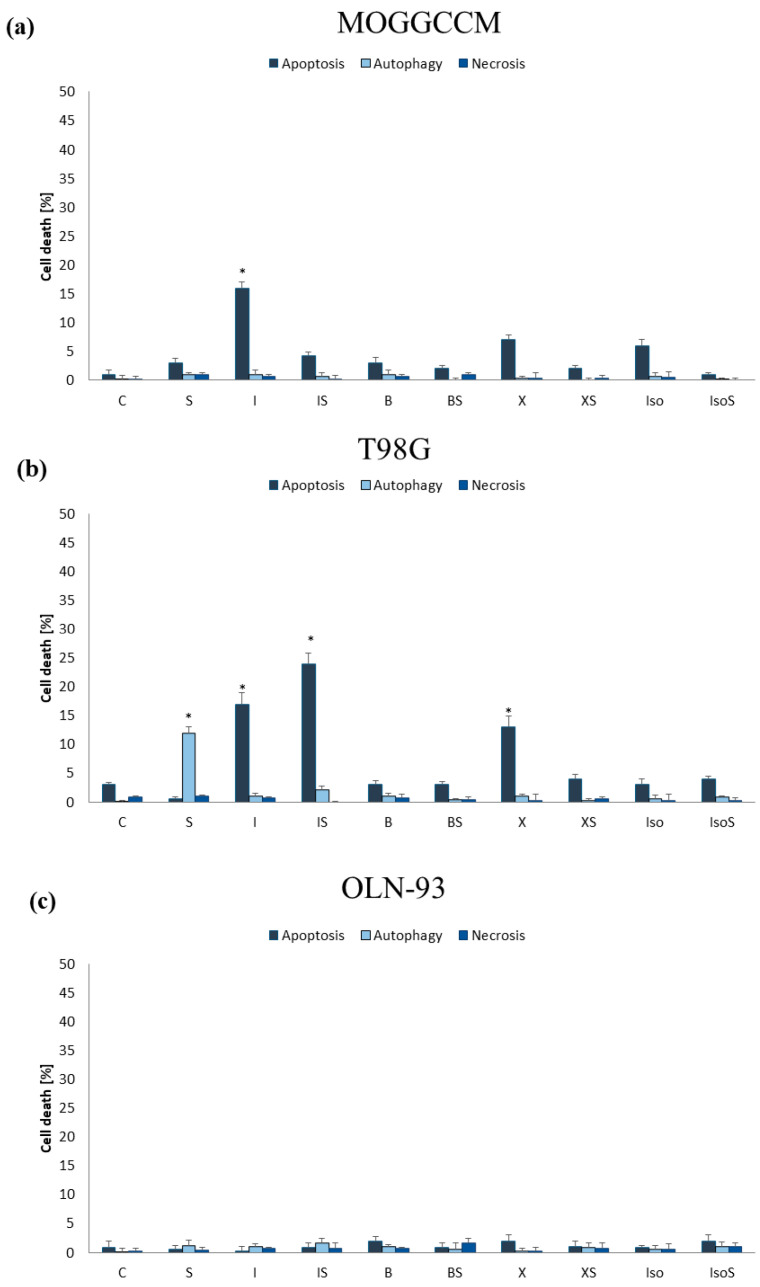
Evaluation of level of cell death observed in MOGGCCM (**a**), T98G (**b**), and OLN-93 (**c**) cells after 24 h incubation with bergapten (B), xanthotoxin (X), imperatorin (I), isoimperatorin (IS), and sorafenib (S) in single or in combination. * *p* < 0.05 compared to control.

**Figure 6 ijms-25-00759-f006:**
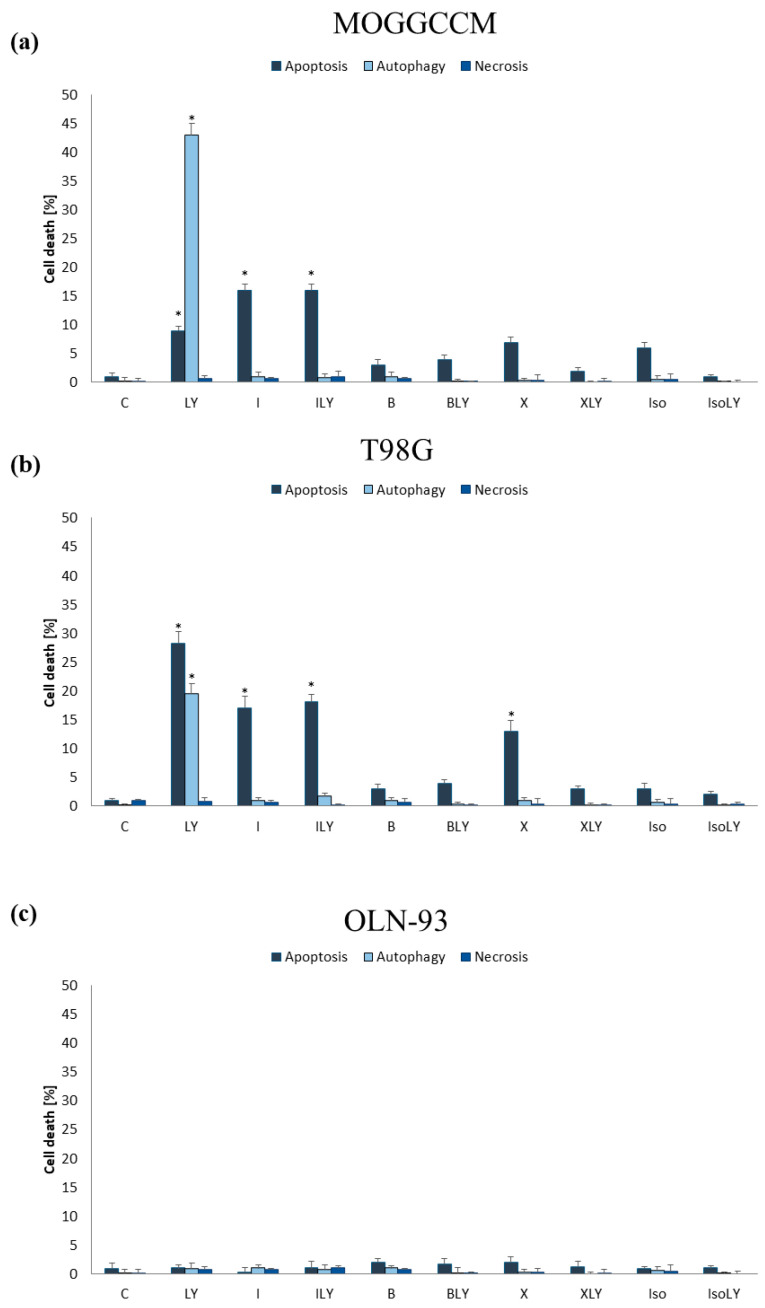
Evaluation of level of cell death observed in MOGGCCM (**a**), T98G (**b**), and OLN-93 (**c**) cells after 24 h incubation with bergapten (B), xanthotoxin (X), imperatorin (I), isoimperatorin (IS), and LY294002 (LY) in single or in combination. * *p* < 0.05 compared to control.

**Figure 7 ijms-25-00759-f007:**
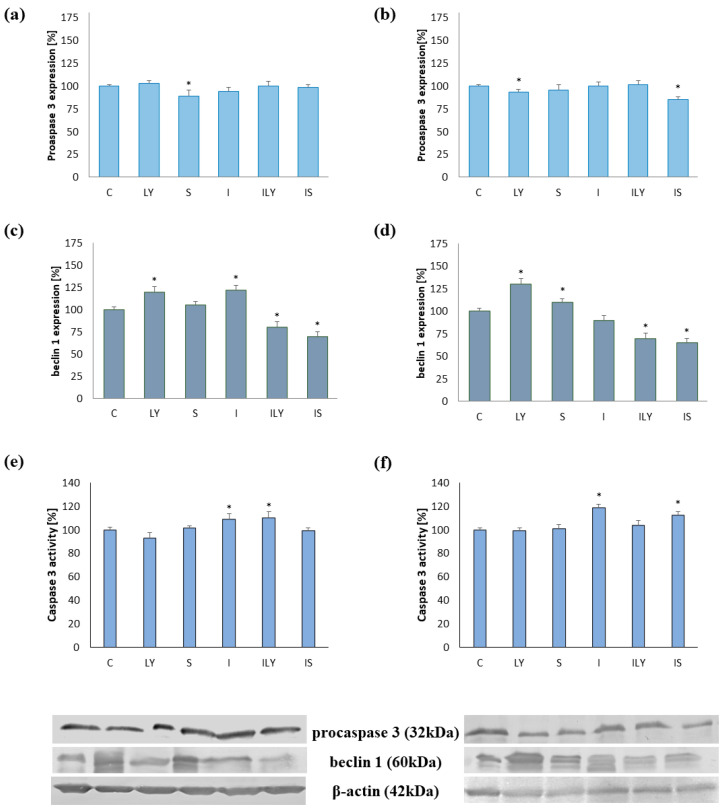
Evaluation of procaspase 3 (**a**,**b**), Beclin 1 (**c**,**d**) level, and activity of caspase 3 in MOGGCCM (**a**,**c**,**e**), T98G (**b**,**d**,**f**) cells after 24 h incubation with imperatorin (I), sorafenib (S), and LY294002 (LY) in single or in combination. * *p* < 0.05 compared to control; whole Western blot membranes attached as Appendix A.

**Figure 8 ijms-25-00759-f008:**
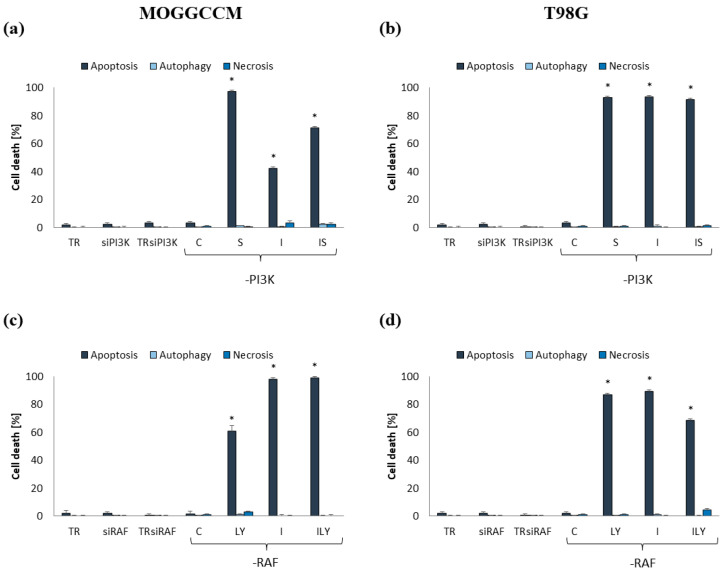
Level of cell death in MOGGCCM and T98G cells transfected with siPI3K (**a**,**b**), and siRAF-1 (**c**,**d**) after 24 h incubation with imperatorin (I), LY294002 (LY), and/or sorafenib (S) in single or in combination. C—control cells; TR—transfection reagent; * *p* < 0.05 compared to control; photos of transfection efficiency attached as Appendix A.

**Figure 9 ijms-25-00759-f009:**
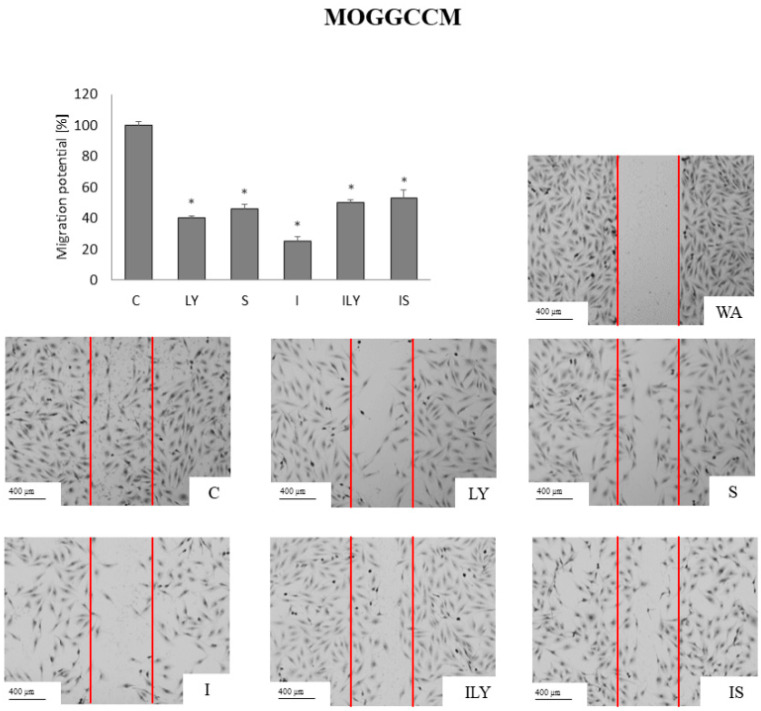
Migration potential of the MOGGCCM cells upon imperatorin (I) and sorafenib (S) or LY294002 (LY) presented as the percent of cells within the woud (WA; red lines on the figures). C-control; * *p* < 0.05.

**Figure 10 ijms-25-00759-f010:**
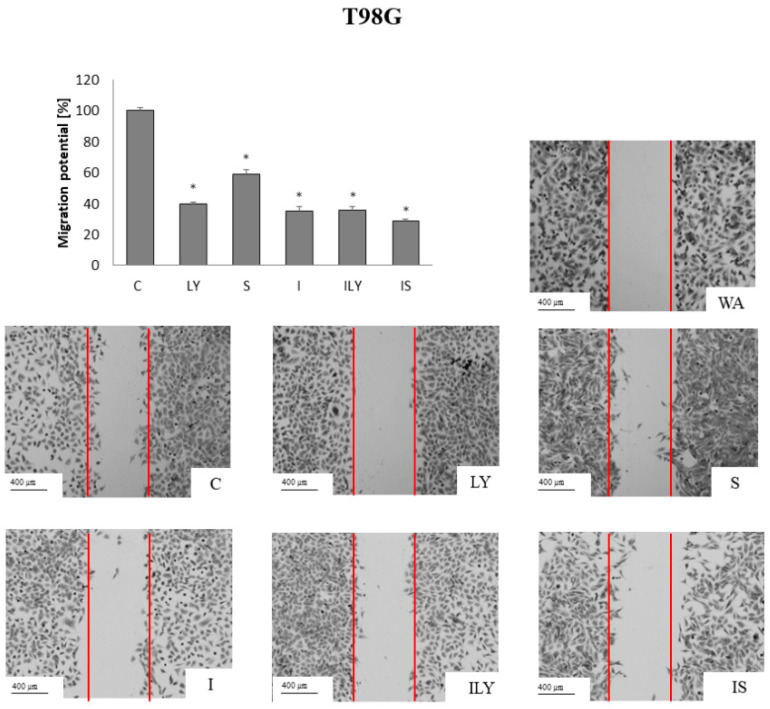
Migration potential of the T98G cells upon imperatorin (I) and sorafenib (S) or LY294002 (LY) presented as the percent of cells within the woud (WA; red lines on the figures). C-control; * *p* < 0.05.

## Data Availability

The data presented in this study are available from the corresponding author on request.

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
