# Peer review of "Furanocoumarins as Enhancers of Antitumor Potential of Sorafenib and LY294002 toward Human Glioma Cells In Vitro"

_ijms, 2024, doi:10.3390/ijms25020759_

Round 1

Reviewer 1 Report

Comments and Suggestions for Authors

The manuscript describes data on usage of different forms of furanocoumarins as an enhancers of antitumor potential of sorafenib and LY294002 in terms of looking for a perscepcives of human gliomas treatment.

The study concerns the most common tumors of the central nervous system constituting approximately 30-40% of intracranial tumors therefore it provides relevant data for anticancer therapy.

The study I my opinion is well designed and all results support assumed conclusions.

However, few points should be revised or explain:

M&M, line 389. Mixture of DMEM and F-12 mediums suplemented with FBS was used for OLN-93 cell line growth. Was this mixture  suplemented with antibiotics as for MOGGCCM or T98G cell line?

Results, 2.1. The authors present data of programmed cell death by different forms of furanocoumarins. In the text there are described concentration of: 50, 100, 150, 200 and 250uM whereas figures 2 and 3 present data for another final concentrations of furacoumarins... Which are correct? Please, check it again. I suggest also to add how was a final concentration of furanocumarins determined to use in the study.

2.2. and 2.3. Could authors precise the concentration of used chemicals i.e. furanocoumarins and sorafenib?

Fig. 5a for IS group statictics is remarked as + - what does it mean? Should it be *?

When it comes to blocking of Raf kinase... which kind of kinase (Raf-1, A-RAF or B-RAF) was it? Did authors use control siRNA?

Fig. 8. Please explain what does mean TR?

Fig.9 and Fig.10. plots contains K groups... should it be C?

Minor points:

In few cases MOGGCCM cells are written uncorrectly: MOGCCM. Please check it carefully

line 21: should be Hoechst 33342

line 91: double involved/involving - sentence should be reworded

line 428: should be caspase 3

Reviewer 2 Report

Comments and Suggestions for Authors

Dear Authors,

I have read this MS ‘Furanocoumarins as an enhancers of antitumor potential of sorafenib and LY294002 toward human glioma cells in vitro’, MS is interesting and I have below suggestions.

1.      Introduction section lacks rigor in literature review. Please see below recent review, Ahmed, S., Khan, H., Aschner, M., Mirzae, H., Küpeli Akkol, E. and Capasso, R., 2020. Anticancer potential of furanocoumarins: mechanistic and therapeutic aspects. International Journal of Molecular Sciences, 21(16), p.5622. You need to enhance this section.

2.      Please add a schematic better to explain the therapeutic mechanism underlying.

3.      All graphs needs to be plotted with Statistical Softwares such as GraphPad Prism. If you don’t have it, it offers trail period. Or remove background lines in current graphs.

4.      Use different color schemes in graph to easily differentiate them.

5.      Line 240, images aren’t sharp. Please localize the changes in morphologies via making circles in Red in images of cells herein.

6.      Line 397-400, please double check what it has been written. You need to mention why?

How many controls? What were trials you repeated?

7.      The experiments lacks SEM (std error of means), write clearly n=?

Comments on the Quality of English Language

NA

Round 2

Reviewer 2 Report

Comments and Suggestions for Authors

Accepted